# Comparing the effectiveness of negative-pressure barrier devices in providing air clearance to prevent aerosol transmission

Tzu-Yao Hung[1,2,3,4‡], Wei-Lun Chen[1‡], Yung-Cheng Su[5,6], Chih-Chieh Wu[1], Tzu-Yao Chueh[7], Hsin-Ling Chen[1☯*], Shih-Cheng Hu[7☯*], Tee Lin[7☯*]

1 Department of Emergency Medicine, Zhong-Xing branch, Taipei City Hospital, Taipei City, Taiwan, 2 Faculty of Medicine, National Yang-Ming Chiao Tung University, Tainan City, Taiwan, 3 School of Medicine, College of Medicine, Fu Jen Catholic University, New Taipei, Taiwan, 4 CrazyatLAB (Critical Airway Training Laboratory), Taipei City, Taiwan, 5 School of Medicine, Tzu Chi University, Hualien County, Taiwan, 6 Department of Emergency, Dalin Tzu Chi Hospital, Buddhist Tzu Chi Medical Foundation, Chiayi County, Taiwan, 7 Department of Energy and Refrigerating Air-conditioning Engineering, National Taipei University of Technology, Taipei, Taiwan

☯ These authors contributed equally to this work.
‡ Contributed equally to this work and are considered co-first authors
* taipeicityerprogram@gmail.com (HLC); f10870@ntut.edu.tw (SCH); whitelin1112@gmail.com (TL)

**Data Availability Statement:** All relevant data are within the paper and its Supporting Information files.

## Abstract

### Purpose

To investigate the effectiveness of aerosol clearance using an aerosol box, aerosol bag, wall suction, and a high-efficiency particulate air (HEPA) filter evacuator to prevent aerosol transmission.

### Methods

The flow field was visualized using three protective device settings (an aerosol box, and an aerosol bag with and without sealed working channels) and four suction settings (no suction, wall suction, and a HEPA filter evacuator at flow rates of 415 liters per minute [LPM] and 530 LPM). All 12 subgroups were compared with a no intervention group. The primary outcome, aerosol concentration, was measured at the head, trunk, and foot of a mannequin.

### Results

The mean aerosol concentration was reduced at the head ($p < 0.001$) but increased at the feet ($p = 0.005$) with an aerosol box compared with no intervention. Non-sealed aerosol bags increased exposure at the head and trunk (both, $p < 0.001$). Sealed aerosol bags reduced aerosol concentration at the head, trunk, and foot of the mannequin ($p < 0.001$). A sealed aerosol bag alone, with wall suction, or with a HEPA filter evacuator reduced the aerosol concentration at the head by 7.15%, 36.61%, and 84.70%, respectively (99.9% confidence interval [CI]: -4.51–18.81, 27.48–45.73, and 78.99–90.40); trunk by 70.95%, 73.99%, and 91.59%, respectively (99.9% CI: 59.83–82.07, 52.64–95.33, and 87.51–

**Funding:** This study was funded by the department of health, Taipei City Government, Taiwan. The funders had no role in study design, data collection and analysis, decision to publish, or preparation of the manuscript.

**Competing interests:** The authors have declared that no competing interests exist.

95.66); and feet by 69.16%, 75.57%, and 92.30%, respectively (99.9% CI: 63.18–75.15, 69.76–81.37, and 88.18–96.42), compared with an aerosol box alone.

## Conclusions

As aerosols spread, an airtight container with sealed working channels is effective when combined with suction devices.

## Introduction

The highest concentrations of severe acute respiratory syndrome-coronavirus-2 (SARS-CoV-2) are found in the saliva, sputum, and upper airway secretions [1]. Coronavirus disease (COVID-19) may spread through small droplets or aerosols [2–8]. Compared with droplets, aerosols spread more easily and also remain in the air for a longer period of time. Aerosols can be generated through coughing, talking, sneezing, breathing, during oxygenation using a high-flow facility, before or during the intubation process, and when managing a deteriorated airway in patients with COVID-19 [2,4–10]. To date, there have been no clinical experiments regarding SARS-CoV-2 aerosol transmission; however, animal models have provided direct evidence that aerosols are an important means of transmission [11].

Several guidelines have recommended the routine use of personal protective equipment (PPE) during high-risk procedures, such as tracheal intubation [6,7,12,13]. Furthermore, novel barrier equipment, such as aerosol boxes (AB) and disposable plastic aerosol bags, have been developed to reduce droplet spillage and transmission risk [14–20]. AB (Taiwan box) is a pioneer public-shared barrier design for effectively preventing droplet spillage [18,21]. Another novel method involves a disposable plastic aerosol bag clipped by strings attached to the surgical lamp in the resuscitation room. This creates a negative-pressure barrier with the wall suction and can be used to reduce the risk of fomite transmission when considering the possibility of inadequate disinfection [19].

To accelerate viral concentration clearance in the air, wall suction is an easily accessible method that can be used to create negative pressure, and this may be useful in combination with protective barriers [19,22]. The high-efficiency particulate air (HEPA) filtering evacuator is another accessible device that is widely used in dermatological interventions such as cryotherapy of pathological lesions infected with human papillomavirus. Furthermore, the HEPA filtering evacuator has been used for pathogens smaller than SARS-CoV-2.

Daily human activity also leads to the production of bioaerosols: breathing (<0.8–2μm), speaking (<0.8–7μm, 16–125μm), shouting (0.5–10μm), coughing (0.62–15.9μm, 40–125μm), and sneezing (7–125μm) [9]. This study aimed to investigate three different devices (an AB and an aerosol bag with and without sealed working channels) in combination with or without suction systems for aerosol protection via flow field visualization. We also aimed to investigate the detection of aerosol exposure during tracheal intubation in order to identify the best negative-pressure barrier system for the protection of healthcare workers from aerosol-generating procedures.

## Methods

### Study design and setting

This *in situ* study was performed in a negative-pressure resuscitation room, with 12 air changes per hour, located at Taipei City Hospital, Zhong-Xing branch, a metropolitan teaching and designated COVID-19 treatment hospital in Taiwan.

The background flow of the resuscitation room runs from the top of the space to four vents at the bottom of each corner and occurs in a downward direction. A simulated mannequin (Laerdal® Airway Management Trainer, Laerdal, New York, USA) was positioned in an inclined head-up 30˚ position, and the trachea was connected to a three-dimensional printed ventilator (Massachusetts Institute of Technology Emergency Ventilator, Massachusetts, USA [23]) and a smoke particle generator (MPL-I003, Tong-Da, Tainan, Taiwan) to create visible smoke (atomized glycerol) from the mouth of the mannequin (Fig 1A). The breathing cycle was set to a fixed rate of 25 times per minute and created a minute ventilation of approximately 10 liters per minute (LPM) to simulate a tachypneic patient. Large-area particle image velocimetry (PIV) was applied to analyze the dynamic flow field of the air during the breathing cycle. A high-sensitivity camera (ORCA-Flash 4.0 v2 digital CMOS camera, Hamamatsu Co., Hamamatsu, Japan) was set in a vertical direction 2 m away from the head of the mannequin to record the light source scattered by the tracer gas of glycerol after being irradiated by the green laser.

To investigate the concentration of the contaminated aerosol, the glycerol tracer gas was then replaced with polyalphaolefin (PAO) (with a diameter of 0.5–0.7 μm). Three spots around the mannequin (head, trunk, and foot) were assessed with a light-scattering photometer with a sampling rate of 28.3 LPM in 180 s intervals (Fig 1B and 1D) to evaluate the aerosol exposure of healthcare workers. Between each setting, detection only started after the concentration of PAO dropped to less than 0.005% (background level).

## Interventions

Two novel protective devices were investigated, an AB and a plastic disposable aerosol bag (PDAB) with and without two sealed working channels (three protective device settings). The AB was a box-like shield with dimensions of 50 cm × 35 cm × 55 cm with no rear plane; two working channels on the front and one working channel was attached on each side of the lateral plane for airway management (Fig 1C and 1D). A PDAB (60 cm × 70 cm) was attached to the surgical light in the resuscitation room and was sufficient to cover the head, shoulder, and the upper chest of the mannequin (Fig 1E).

The following two suction devices were evaluated during the study: a wall suction (3-Stages Analogue Vacuum Regulator, maximum flow rate 36 LPM, Pacific Hospital Supply Co., Taipei, Taiwan) and an evacuator with a HEPA filter (Surgifresh Mini TURBO Smoke Evacuators, with a minimum flow rate of 415 LPM, and a maximum of 530 LPM, Dynamic Medical Technologies, Taipei, Taiwan). The suction devices were placed on the chin of the mannequin.

Two protective devices with three settings (AB, non-sealed working channels PDAB, sealed working channels PDAB) and two suction devices with four settings (no suction device, wall suction at a flow rate of 36 LPM, and HEPA filter evacuator at flow rates of 415 LPM and 530 LPM) created 12 (3 × 4) possible conditions, all of which were compared with the no intervention condition (total 13 subgroups).

## Measurements

The flow field visualization was recorded by a high-sensitivity camera in both sagittal and coronal views. The background flow field was recorded first. A total of 13 different settings were recorded: no intervention, AB, AB with wall suction at a flow rate of 36 LPM, and AB with HEPA filter evacuator (HE) at a flow rate of 415 LPM and 530 LPM; DPAB without sealed working channels (DPAB-NS), DPAB-NS with wall suction, and DPAB-NS with HE at 415 LPM and 530 LPM; DPAB with seal (DPAB-S) and DPAB-S with wall suction;

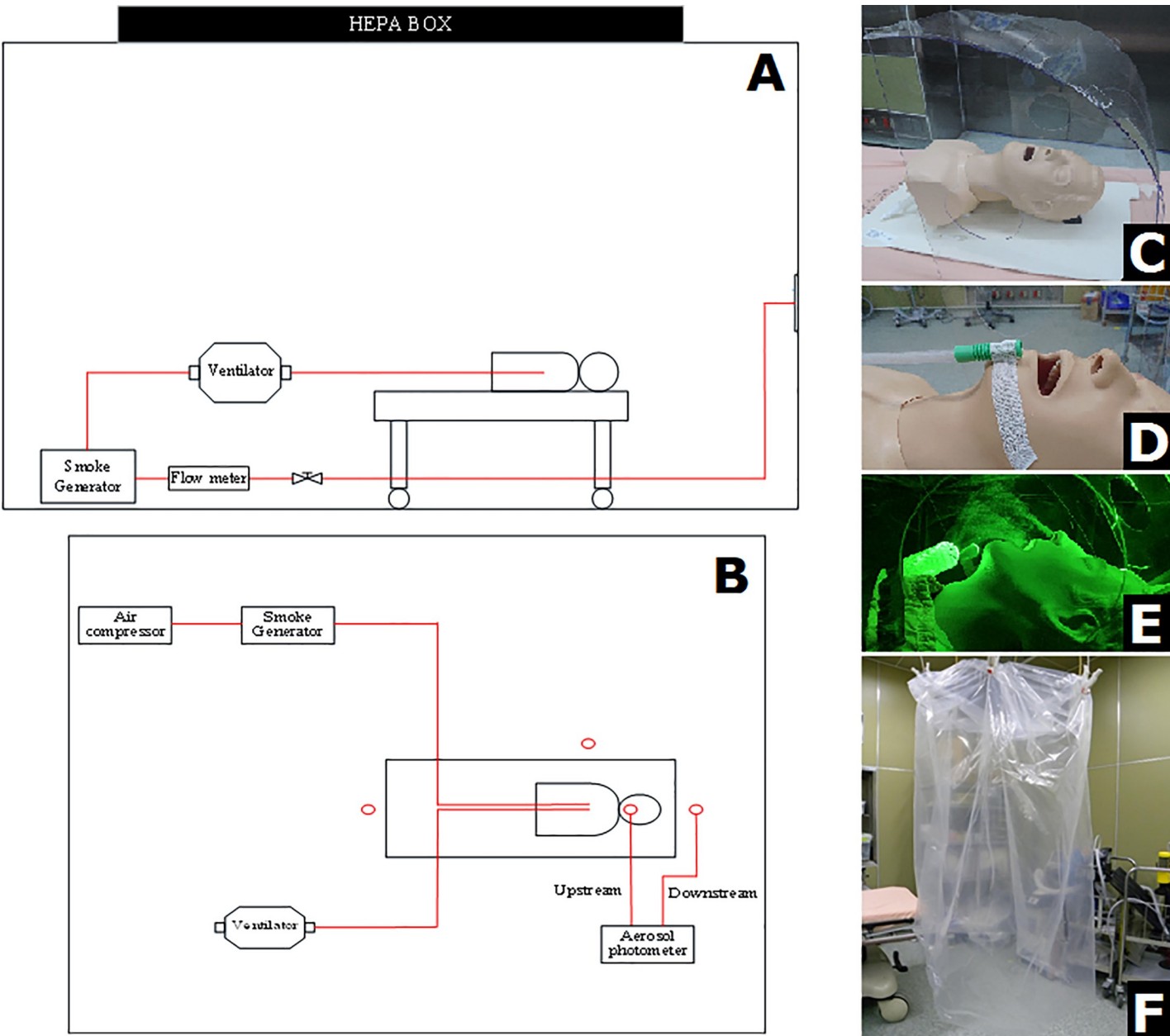

**Fig 1.** Two novel protective devices: A. Experimental setting. The ventilator was connected to a smoke generator and air supplement system; then, the ventilator was set to 25 breathing cycles per minute (10 L/min) and attached to the mannequin. A green laser was ejected from the foot and trunk of the mannequin to demonstrate the aerosol in two vertical dimensions. B. Aerosol concentration experimental setting. The ventilator was connected with the smoke generator and the mannequin, and the particles were pumped out during the breathing cycle, simulating aerosols. The upstream detector of the aerosol photometer was placed by the mouth of the mannequin, while the downstream detector was placed in the three spots (head, trunk, and foot of the mannequin). C. Aerosol box (side view), a shield-like device, covers the front, top, and two lateral sides; the rear side is left open for equipment transfer. There were two working channels in the front of the box, and two at each lateral side of the box for airway management. D. The wall suction was attached at the chin of the mannequin using an elastic adhesive tape. E. The high-efficiency particulate air (HEPA) evacuator was equipped with a metal stand that can clip and fix the suction tube upright just above the chin of the mannequin. F. Disposable transparent plastic bag (aerosol bag).

and DPAB-S with HE at 415 LPM and 530 LPM. The primary outcome was PAO concentration at the head, trunk, and foot of the mannequin at an interval of 180 s among 13 subgroups.

### Statistical analysis

Student t-tests were used to evaluate the percentage differences between each protective equipment setting. SAS statistical package version 9.4 (SAS Institute, Inc., North Carolina, USA) and STATA version 15.1 (StataCorp, Texas, USA) were used for all data analysis.

Two-tailed *p*-values <0.001 were considered to be statistically significant.

The sample size calculation for this experiment was based on an initial pilot experiment showing that an average concentration of 600 ppm was detected at the head side when no protective equipment was used. To detect a mean percentage difference of 20% from the baseline level, a two-sided significance level of 0.1%, and power of 80%, an estimated 394 participants were needed per device tested. We have accounted for multiple testing issues using Bonferroni correction method. There was a control group; further, 12 groups with 3 different detecting positions for estimation were also present, bringing the total to 36 comparisons. We adjusted the p-value of significance to 0.001 because of the large number of pairwise comparisons. As this was a simulated mannequin study and involved no human participants, ethical approval was not required.

## Results

Particle visualization images from the sagittal and coronal views showed that aerosol particles can escape from the working channels of AB and DPAB-NS. The aerosols moved along the top of the AB to the foot of the mannequin (Fig 2A). However, DPAB-S had a better ability to confine the aerosol particles without visible aerosol escape.

With regard to suction devices, wall suction decreased the escape of the aerosol particles when used in combination with AB and DPAB-NS. However, aerosol escape was still observed

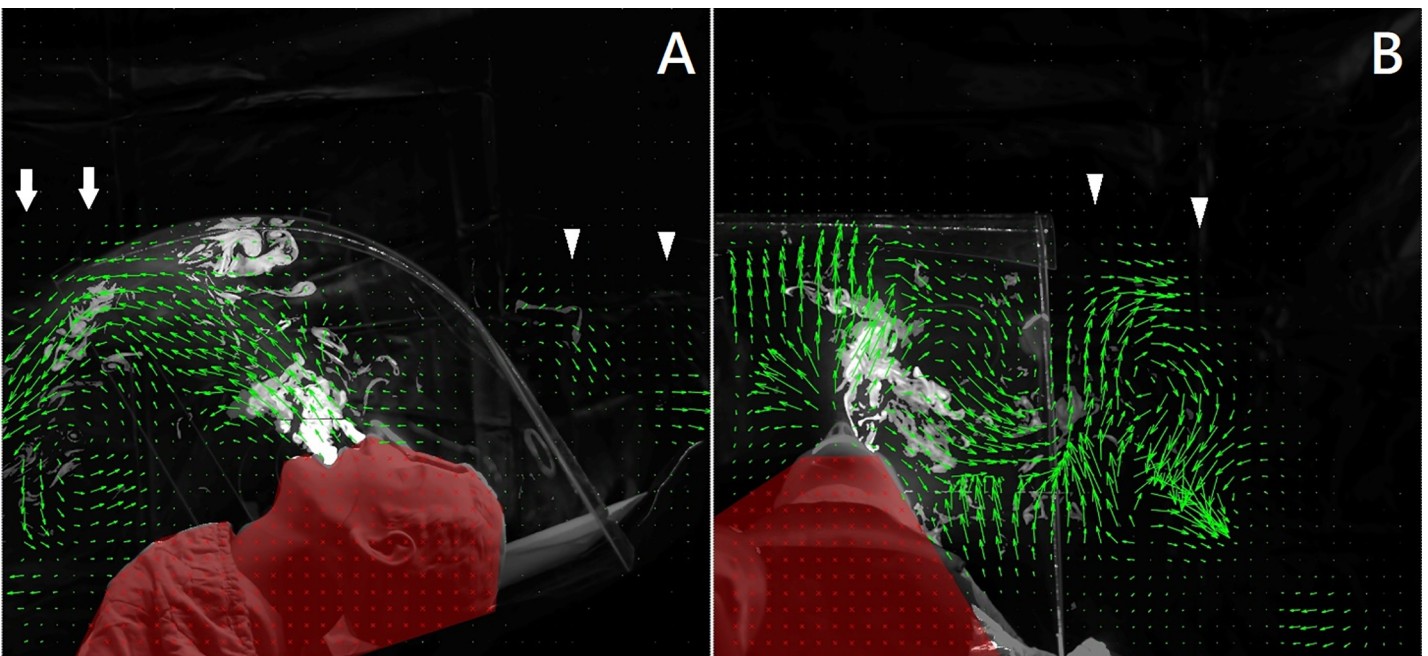

**Fig 2. Particle visualization images in both the sagittal and coronal views.** A. Sagittal view of the aerosol box containing aerosols; aerosol movement is visualized with green vector arrows in the visualized flow field. The white arrows indicate the aerosols moving to the foot of the mannequin. The white arrowheads indicate leakage of aerosols from the front non-sealed working channels. B. Coronal view of the visualized flow field. The white arrowheads indicate aerosol escape from the side non-sealed working channels.

in the coronal view of AB. The minimum (415 LPM) or maximum (530 LPM) HE flow rate decreased the dispersion of the aerosols, and there was no visualized escape. DPAB-S showed no visualized aerosol escape with and without all suction systems.

Compared to baseline concentrations without intervention, AB decreased the aerosol exposure at the head of the mannequin, but this was increased at the foot of the mannequin (p = 0.05). DPAB-NS decreased exposure at the foot of the mannequin; however, exposure at the head and trunk of the mannequin markedly increased. DPAB-S had lowest exposure among the barriers at all detection sites of all the cases without suction devices (Table 1).

In combination with wall suction at a flow rate of 36 LPM, aerosol exposure decreased significantly (p<0.001) with AB, DPAB-NS, and DPAB-S. However, AB still showed the highest exposure of all the barriers at the foot of the mannequin; the highest aerosol concentration at the trunk of the mannequin was found with DPAB-NS. Of all the barrier devices, DPAB-S showed the lowest aerosol concentration at all three detection sites (Table 1).

In combination with HE at a flow rate of 415 and 530 LPM, aerosol exposure was very low with all three barrier devices. Overall, the DPAB-S subgroup showed the lowest concentration compared with other subgroups (Table 1).

With suction, AB was found to be more protective for workers at the head and the trunk of the mannequin but worse for those at the foot of the mannequin, compared with the protective effect of DPAB-NS (Table 1). However, without the suction system, DPAB-S was the best barrier device at all three detection sites (all, $p < 0.001$).

Comparing the three devices (AB, DPAB-NS, and DPAB-S) in combination with the three suction settings (wall suction at 36 LPM, HE at 415 LPM, and HE at 530 LPM), DPAB-S showed a better protective effect from the aerosols than AB (Table 2) at all three detection sites (head, trunk, and foot) of the mannequin ($p < 0.001$). For non-sealed barriers, AB with wall suction was protective at the trunk of the mannequin but not at the foot of the mannequin, as compared to DPAB-NS ($p < 0.001$). With HE, there were no differences between DPAB-NS and AB at any of the three detection sites (Table 2).

## Discussion

The COVID-19 pandemic has driven clinicians to develop devices that protect healthcare workers from nosocomial infections during tracheal intubation [14–21]. The Taiwan box is a pioneering public-shared design [21]. Canelli et al. found that such an AB can prevent droplet spillage and splashes while a patient is coughing [18]. We also fabricated a device with a single-use plastic bag to create a negative-pressure barrier with the wall suction (published on April 1st, 2020). Any other such sufficiently large transparent plastic bag can be used for this negative barrier system (Figs 1E and 3A–3D) [19]. However, the effectiveness of these protective devices is controversial; Begley et al. found that the AB may increase intubation time and decrease the first-pass success [15].

Flow visualization using the laser PIV technique [24,25], found that the aerosol flowed upward and then followed the top curve of AB to the foot of the mannequin, which could be a health hazard if the health workers at the foot wore insufficient PPE (Fig 2A, S1 Video). Furthermore, the flow can escape significantly from the non-sealed working channels with time, thus being even more dangerous (Fig 4A–4F). When the plastic aerosol bag was sealed with a working channel to form an airtight container, the aerosols were confined better, without significant leakage. Sealing of the working channels can improve aerosol entrapment efficiency. When using a wall suction at 36 LPM, the clearance of aerosols was insufficient; clearance improved with time if this was combined with the AB (Fig 4D–4F blue line) and DPAB-NS (Fig 4D–4F pink line). With HE at flow rates of 415 and 530 LPM, aerosol clearance was fast,

**Table 1. Aerosol concentration of the control group and 12 different subgroup settings at the head, trunk, and foot of the mannequin.**

| | N | Group | | Suction Device (LPM) | | Mean (ppm) | p-value | 99.9% CI | | |
|---|---|---|---|---|---|---|---|---|---|---|
| | **499** | **No intervention** | | | | **662.03** | | **565.83** | ~ | **758.24** |
| H | 495 | Aerosol Box | | | | 483.80 | <0.001 | 450.36 | ~ | 517.24 |
| | 490 | Aerosol Box | + | Wall suction | 36 | 277.36 | <0.001 | 256.63 | ~ | 298.08 |
| | 473 | Aerosol Box | + | HEPA evacuator | 415 | 47.27 | <0.001 | 44.74 | ~ | 49.80 |
| E | 487 | Aerosol Box | + | HEPA evacuator | 530 | 26.07 | <0.001 | 24.88 | ~ | 27.27 |
| | 498 | Non-sealed aerosol bag | | | | 1214.25 | <0.001 | 92.36 | ~ | 1500.14 |
| | 510 | Non-sealed aerosol bag | + | Wall suction | 36 | 266.40 | <0.001 | 224.00 | ~ | 308.80 |
| A | 471 | Non-sealed aerosol bag | + | HEPA evacuator | 415 | 47.27 | <0.001 | 44.73 | ~ | 49.80 |
| | 488 | Non-sealed aerosol bag | + | HEPA evacuator | 530 | 26.01 | <0.001 | 24.82 | ~ | 27.21 |
| | 489 | Sealed Aerosol bag | | | | 449.20 | <0.001 | 403.41 | ~ | 494.99 |
| D | 488 | Sealed Aerosol bag | + | Wall suction | 36 | 175.83 | <0.001 | 161.22 | ~ | 190.44 |
| | 480 | Sealed Aerosol bag | + | HEPA evacuator | 415 | 7.23 | <0.001 | 6.23 | ~ | 8.23 |
| | 486 | Sealed Aerosol bag | + | HEPA evacuator | 530 | 5.19 | <0.001 | 4.57 | ~ | 5.82 |
| T | **493** | **No intervention** | | | | **345.83** | | **321.76** | ~ | **369.89** |
| | 508 | Aerosol Box | | | | 410.21 | <0.001 | 366.91 | ~ | 453.51 |
| R | 494 | Aerosol Box | + | Wall suction | 36 | 241.08 | <0.001 | 190.26 | ~ | 291.90 |
| | 478 | Aerosol Box | + | HEPA evacuator | 415 | 38.91 | <0.001 | 37.39 | ~ | 40.42 |
| | 477 | Aerosol Box | + | HEPA evacuator | 530 | 23.56 | <0.001 | 22.43 | ~ | 24.68 |
| U | 504 | Non-sealed aerosol bag | | | | 7705.68 | <0.001 | 5651.59 | ~ | 9759.76 |
| | 499 | Non-sealed aerosol bag | + | Wall suction | 36 | 855.73 | <0.001 | 513.26 | ~ | 1198.20 |
| | 474 | Non-sealed aerosol bag | + | HEPA evacuator | 415 | 38.79 | <0.001 | 37.34 | ~ | 40.23 |
| N | 475 | Non-sealed aerosol bag | + | HEPA evacuator | 530 | 23.53 | <0.001 | 22.41 | ~ | 24.66 |
| | 484 | Sealed Aerosol bag | | | | 119.17 | <0.001 | 107.69 | ~ | 130.66 |
| | 483 | Sealed Aerosol bag | + | Wall suction | 36 | 62.71 | <0.001 | 57.91 | ~ | 67.52 |
| K | 494 | Sealed Aerosol bag | + | HEPA evacuator | 415 | 3.27 | <0.001 | 2.73 | ~ | 3.81 |
| | 495 | Sealed Aerosol bag | + | HEPA evacuator | 530 | 2.83 | <0.001 | 2.16 | ~ | 3.51 |
| | **494** | **No intervention** | | | | **693.07** | | **642.78** | ~ | **743.36** |
| F | 504 | Aerosol Box | | | | 748.31 | 0.005 | 707.93 | ~ | 788.69 |
| | 499 | Aerosol Box | + | Wall suction | 36 | 450.01 | <0.001 | 425.50 | ~ | 474.52 |
| | 472 | Aerosol Box | + | HEPA evacuator | 415 | 26.78 | <0.001 | 25.71 | ~ | 27.85 |
| O | 487 | Aerosol Box | + | HEPA evacuator | 530 | 28.17 | <0.001 | 27.10 | ~ | 29.23 |
| | 500 | Non-sealed aerosol bag | | | | 462.41 | <0.001 | 391.65 | ~ | 533.18 |
| | 506 | Non-sealed aerosol bag | + | Wall suction | 36 | 144.23 | <0.001 | 130.72 | ~ | 157.73 |
| O | 470 | Non-sealed aerosol bag | + | HEPA evacuator | 415 | 26.74 | <0.001 | 25.67 | ~ | 27.81 |
| | 477 | Non-sealed aerosol bag | + | HEPA evacuator | 530 | 28.24 | <0.001 | 27.16 | ~ | 29.31 |
| | 487 | Sealed Aerosol bag | | | | 230.74 | <0.001 | 212.20 | ~ | 249.29 |
| T | 486 | Sealed Aerosol bag | + | Wall suction | 36 | 109.95 | <0.001 | 101.53 | ~ | 118.37 |
| | 495 | Sealed Aerosol bag | + | HEPA evacuator | 415 | 2.06 | <0.001 | 1.70 | ~ | 2.42 |
| | 495 | Sealed Aerosol bag | + | HEPA evacuator | 530 | 1.77 | <0.001 | 1.45 | ~ | 2.09 |

HEPA, high-efficiency particulate air; LPM, liters per minute.

without visible aerosol flow escaping from the rear end and working channels of AB; the plastic aerosol bag also showed similar results.

Simpson et al. investigated the function of four protective devices using aerosols (AB, sealed box, and vertical and horizontal drapes) and found that AB and horizontal and vertical plastic drapes were of no use and might increase the aerosol contamination in the intubator [22]. To

**Table 2. Non-sealed and sealed aerosol bags versus aerosol boxes in combination with or without suction devices.**

| | Settings | | | Suction Device | Mean Percentage Difference(%)* | 99.9% Confidence Interval(%) | p-value |
|---|---|---|---|---|---|---|---|
| | Aerosol Box | vs. | Non-sealed Aerosol Bag | None | -150.98 | -210.47 to -91.49 | <0.001 |
| H | Aerosol Box | vs. | Sealed Aerosol Bag | None | 7.15 | -4.51~ 18.81 | 0.044 |
| E | Aerosol Box | vs. | Non-sealed Aerosol Bag | Wall Suction | 3.95 | -13.23~ 21.13 | 0.424 |
| A | Aerosol Box | vs. | Sealed Aerosol Bag | Wall Suction | 36.61 | 27.48~ 45.73 | <0.001 |
| D | Aerosol Box | vs. | Non-sealed Aerosol Bag | HEPA Evacuator (415 LPM) | 0.01 | -7.55~ 7.57 | 0.996 |
| | Aerosol Box | vs. | Sealed Aerosol Bag | HEPA Evacuator (415 LPM) | 84.70 | 78.99~ 90.40 | <0.001 |
| | Aerosol Box | vs. | Non-sealed Aerosol Bag | None | -1778.48 | -2275.88 to -1281.08 | <0.001 |
| T | Aerosol Box | vs. | Sealed Aerosol Bag | None | 70.95 | 59.83~ 82.07 | <0.001 |
| R | Aerosol Box | vs. | Non-sealed Aerosol Bag | Wall Suction | -254.96 | -398.84 to -111.08 | <0.001 |
| U | Aerosol Box | vs. | Sealed Aerosol Bag | Wall Suction | 73.99 | 52.64~ 95.33 | <0.001 |
| N | Aerosol Box | vs. | Non-sealed Aerosol Bag | HEPA Evacuator (415 LPM) | 0.30 | -5.07 to 5.67 | 0.854 |
| K | Aerosol Box | vs. | Sealed Aerosol Bag | HEPA Evacuator (415 LPM) | 91.59 | 87.51~ 95.66 | <0.001 |
| | Aerosol Box | vs. | Non-sealed Aerosol Bag | None | 38.21 | 27.37~ 49.04 | <0.001 |
| F | Aerosol Box | vs. | Sealed Aerosol Bag | None | 69.16 | 63.18~ 75.15 | <0.001 |
| O | Aerosol Box | vs. | Non-sealed Aerosol Bag | Wall Suction | 67.95 | 61.77~ 74.13 | <0.001 |
| O | Aerosol Box | vs. | Sealed Aerosol Bag | Wall Suction | 75.57 | 69.76~ 81.37 | <0.001 |
| T | Aerosol Box | vs. | Non-sealed Aerosol Bag | HEPA Evacuator (415 LPM) | 0.15 | -5.47 to 5.78 | 0.928 |
| | Aerosol Box | vs. | Sealed Aerosol Bag | HEPA Evacuator (415 LPM) | 92.30 | 88.18~ 96.42 | <0.001 |

HEPA, high-efficiency particulate air; LPM, liters per minute.

further investigate aerosol exposure, we chose three sites to record aerosol concentration: the intubator (head), first assistant (trunk), and second assistant (foot). We found that the use of AB can lead to reduction in intubator exposure. However, the aerosol concentration at the foot of the mannequin was even higher than that at the head of the mannequin (Fig 2A, white arrows, Fig 4A–4C, blue line). With DPAB-NS, a significant amount of aerosol escaped from the working channels near the head and trunk, thus posing a hazard to both the intubator and first assistant (Fig 4A–4E pink line, Table 1). Wall suction with AB significantly reduced aerosols at the head of the mannequin; however, aerosol concentration was markedly higher on the foot side, compared to no intervention; DPAB-NS with wall suction markedly increased the head and trunk-side exposure, only reducing the exposure at the foot, compared to the no

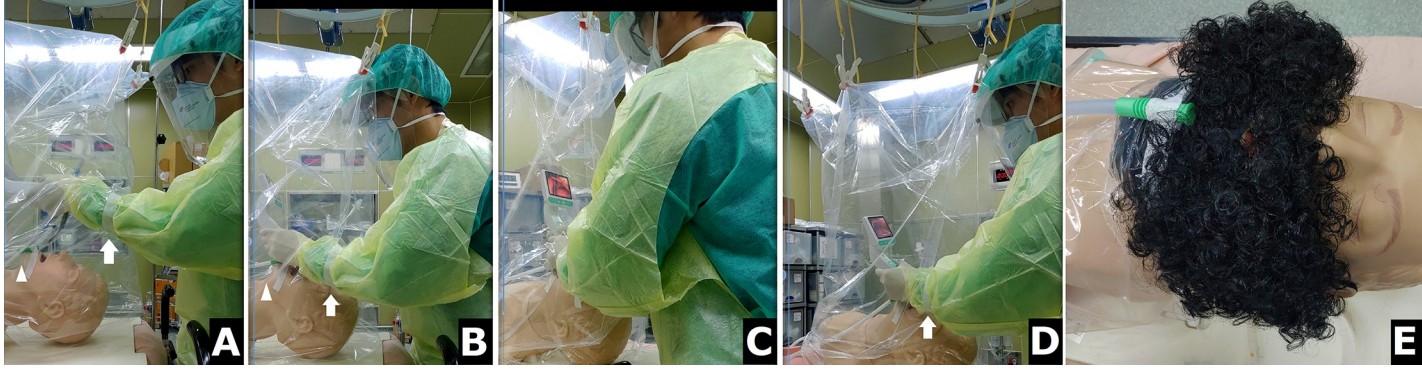

**Fig 3.** A-D. The drill performed with a disposable plastic aerosol bag with sealed working channels (DPAB-S). White arrows indicate the elastic adhesive bandages used to seal the working channels; white arrowheads indicate the suction tube located on the chin of the mannequin. E. When managing patients with a thick beard, a plastic wrap can help fix the suction tube to the patient's chin.

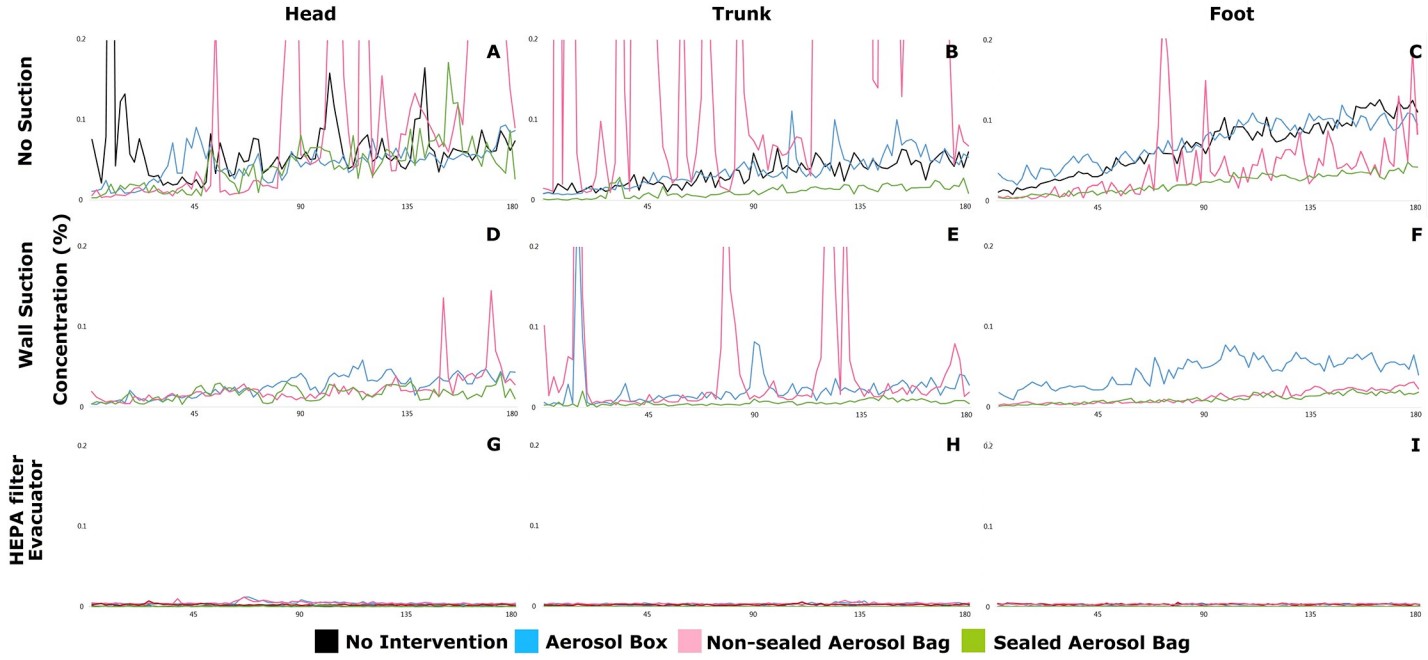

**Fig 4. The aerosol concentration of the 13 different subgroups during a 180 s interval (vertical axis: Concentration, 0–0.2%; horizontal axis: Time, 0–180 s).** Black line: No intervention; blue line: aerosol box; pink line: Non-sealed aerosol bag; green line: Sealed aerosol bag. A, B, and C (upper row) are head, trunk, and foot, with no suction. D, E, and F (middle row) are head, trunk, and foot, with wall suction. G, H, and I (bottom row) are the head, trunk, and foot, with high-efficiency particulate air (HEPA) filter evacuator at 415 liters per minute (LPM) and 530 LPM. Aerosol box (blue line) showed increasing aerosol concentration at the foot of the mannequin with time. Non-sealed aerosol bag (pink line) was hazardous for the healthcare workers at the head and trunk. A sealed aerosol bag (green line) showed better protective effect. With the HEPA filter evacuator, the aerosol concentrations were stable and low constantly (bottom row).

intervention group (Table 1). HE showed sufficient ability for aerosol clearance without leakage with AB and DPAB at all times (Fig 4G–4I).

The working channels of AB and DPAB enable intubation to be performed; however, aerosols were found to escape from these channels, as assessed through flow field visualization and quantification of aerosol concentration (Fig 2, white arrowhead; Fig 4A–4F blue and pink line). Thus, sealing of these channels is crucial for preventing the spread of aerosols and for the protection of healthcare workers; this should, therefore, be recommended. DPAB-S significantly reduced the aerosols at all areas where measurement was performed, with and without suction systems (Table 1). Without the suction system, DPAB-S still reduced aerosol concentrations at the head, trunk, and foot of the mannequin, and this was done to a greater degree than that of AB (Table 2). With the wall suction at a flow rate of 36 LPM at the mouth of the mannequin, DPAB-S reduced aerosol concentrations at the head, trunk, and foot of the mannequin to a greater degree than that of AB. If used with HE at a flow rate of 415 and 530 LPM, DPAB-S reduced the aerosols at the head, trunk, and foot of the mannequin, and to a greater degree than that of AB (Table 2). However, a team-based practice or a drill should be performed to become familiar with the process before using these barriers with sealed working channels during negative-pressure barrier airway management. During our drills, we attached elastic adhesive bandages to seal the working channels with the intubator's forearms (Fig 3A–3D). The flexible nature of DPAB-S allowed the intubator to manage the airway. The assistants need to seal the working channels and activate the suction devices. However, exaggerated movement of the intubator needs to be avoided to minimize the aerosol dispersion. As shown in Fig 3E, a plastic wrap should be applied to fix the wall suction tube to the patient's chin if

managing a patient with a thick beard. A stepwise airway management plan should be practiced.

Wall suction is readily available in different units of hospitals and in ambulances and is easier to use in combination with PPE and a barrier to create a negative-pressure environment. Simpson et al. found that a sealed AB along with wall suction was the only effective setting to reduce aerosol concentration compared to AB without seal and the use of vertical and horizontal drapes [22]. In our study, the wall suction located on the chin of the mannequin (Fig 1D) reduced the concentration of aerosols to some extent; the aerosols were also observed to slightly accumulate and escape from the protective devices over time if the working channels were not sealed (Fig 4D–4F). However, the trend of aerosol concentration was flat and reduced compared to that in the no-suction system subgroups (Fig 4A–4C). In this current study, HE provided sufficient clearance ability, prevented aerosol escape, and increased the safety of protective devices (Table 1). AB is a firm shield that cannot confine aerosols in the same way as DPAB or drapes. However, it is easier to form a closed system and confine the aerosols if the channels are sealed (Tables 1 and 2, Fig 3). The entrapment ability of the protective device and clearing efficiency of the suction system create the best combination of negative-pressure barriers for use during aerosol-generating procedures.

This study has some limitations. First, this was a simulation study and may not represent actual clinical conditions. Minute ventilation was simulated at a fixed rate of 10 LPM, which may not be comparable with human lung physiology. A patient with respiratory failure may present with larger minute ventilation that may be as high as 40–120 LPM. However, the result was still significant, even with a relatively small minute ventilation. Furthermore, intubating patients with COVID-19 in respiratory failure while wearing PPE and using a negative-pressure barrier system requires practice. If untrained, 100–180 s may not be sufficient for the intubation process. The study was also conducted in a negative-pressure room with 12 air changes per hour, and this only meets the minimum requirement of the American Society of Anesthesiologists guidelines for caring for COVID-19 patients. The aerosol concentration may be reduced in a room that has higher air changes per hour. Finally, before applying sealed working channel barriers, team-based practice sessions should be performed to ensure that the fully PPE-equipped intubator and the assistants can cooperate well while working to avoid delays in airway management.

## Conclusions

To facilitate the elimination of the aerosols, wall suction can, to some extent, provide clearing ability; however, the aerosols will still increase within 180 s. HE, at a flow rate of $\geq 415$ LPM, forms a safe negative protective system with both AB and DPAB, and this will be suitable for healthcare workers performing aerosol-generating procedures, especially during pandemics, such as the current COVID-19 outbreak. A negative-pressure barrier consisting of an airtight container and suction devices with sufficient aerosol elimination is effective for aerosol protection.

## Supporting information

**S1 Video. The video demonstrates the aerosol movement to the foot and escape from the working channels of the aerosol box without suction systems.**
(MP4)

## Author Contributions

**Conceptualization:** Tzu-Yao Hung, Wei-Lun Chen, Yung-Cheng Su, Hsin-Ling Chen, Shih-Cheng Hu, Tee Lin.

**Data curation:** Tzu-Yao Hung, Hsin-Ling Chen, Shih-Cheng Hu, Tee Lin.

**Formal analysis:** Tzu-Yao Hung.

**Funding acquisition:** Tzu-Yao Hung, Wei-Lun Chen.

**Investigation:** Tzu-Yao Hung, Wei-Lun Chen, Chih-Chieh Wu, Tzu-Yao Chueh, Hsin-Ling Chen, Shih-Cheng Hu, Tee Lin.

**Methodology:** Tzu-Yao Hung, Wei-Lun Chen, Yung-Cheng Su.

**Project administration:** Tzu-Yao Hung, Chih-Chieh Wu, Tzu-Yao Chueh.

**Resources:** Hsin-Ling Chen, Tee Lin.

**Software:** Yung-Cheng Su, Tee Lin.

**Supervision:** Hsin-Ling Chen, Shih-Cheng Hu, Tee Lin.

**Validation:** Tzu-Yao Hung, Wei-Lun Chen, Yung-Cheng Su, Chih-Chieh Wu, Hsin-Ling Chen, Shih-Cheng Hu.

**Visualization:** Shih-Cheng Hu.

**Writing – original draft:** Tzu-Yao Hung, Wei-Lun Chen.

**Writing – review & editing:** Yung-Cheng Su, Chih-Chieh Wu, Hsin-Ling Chen, Shih-Cheng Hu.

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
