## [Decision Letter · Decision Letter 0]

16 Mar 2021

PONE-D-21-06215

Comparing the effectiveness of negative pressure barrier devices in providing air clearance to prevent aerosol transmission

PLOS ONE

Dear Dr. Chen,

Thank you for submitting your manuscript to PLOS ONE. After careful consideration, we feel that it has merit but does not fully meet PLOS ONE’s publication criteria as it currently stands. Therefore, we invite you to submit a revised version of the manuscript that addresses the points raised during the review process.

It is an interesting research. I received comments and recommendation from one reviewe.Please address these comments and really improve the quality in both English and technical issues.

We look forward to receiving your revised manuscript.

Kind regards,

Jianguo Wang, PhD

Academic Editor

PLOS ONE

Journal Requirements:

'..The study was funded by the Department of Health, Taipei City Government.'

'The funders had no role in study design, data collection and analysis, decision to publish, or preparation of the manuscript.'

3. We note that Figure 3 includes images of an individual.

As per the PLOS ONE policy (http://journals.plos.org/plosone/s/submission-guidelines#loc-human-subjects-research) on papers that include identifying, or potentially identifying, information, the individual(s) or parent(s)/guardian(s) must be informed of the terms of the PLOS open-access (CC-BY) license and provide specific permission for publication of these details under the terms of this license.

Please download the Consent Form for Publication in a PLOS Journal (http://journals.plos.org/plosone/s/file?id=8ce6/plos-consent-form-english.pdf). The signed consent form should not be submitted with the manuscript, but should be securely filed in the individual's case notes.

Please amend the methods section and ethics statement of the manuscript to explicitly state that the patient/participant has provided consent for publication: “The individual in this manuscript has given written informed consent (as outlined in PLOS consent form) to publish these case details”.

Reviewers' comments:

Reviewer's Responses to Questions

**Comments to the Author**

1. Is the manuscript technically sound, and do the data support the conclusions?

Reviewer #1: Yes

2. Has the statistical analysis been performed appropriately and rigorously? 

Reviewer #1: Yes

3. Have the authors made all data underlying the findings in their manuscript fully available?

Reviewer #1: Yes

4. Is the manuscript presented in an intelligible fashion and written in standard English?

Reviewer #1: No

5. Review Comments to the Author

Reviewer #1: This is an interesting article regarding the effectiveness of negative pressure barrier devices in providing air clearance to prevent aerosol transmission.

The topic is promising

I have the following comments:

- The whole manuscript should be edited for english-language usage

- In the introduction the authors must describe the main aim avoiding too detailed descriptions that should be reported in the methods (primary and secondary etc.)

- Is there a study period?

- In the introduction the authors give too many references to describe a concept (2-10). Should be reduced by adding the following guidelines on the use of laparoscopy in surgery (and the connection with aerosolization)

The risk of COVID-19 transmission by laparoscopic smoke may be lower than for laparotomy: a narrative review. Surg Endosc. 2020 Aug;34(8):3298-3305. doi: 10.1007/s00464-020-07652-y

Italian society of colorectal surgery recommendations for good clinical practice in colorectal surgery during the novel coronavirus pandemic. Tech Coloproctol. 2020 Jun;24(6):501-505. doi: 10.1007/s10151-020-02209-6

A Low-cost, Safe, and Effective Method for Smoke Evacuation in Laparoscopic Surgery for Suspected Coronavirus Patients. Ann Surg. 2020 Jul;272(1):e7-e8. doi: 10.1097/SLA.0000000000003965

- Were any patients involved? How many?

- The authors must clarify the methodology of the study

6. PLOS authors have the option to publish the peer review history of their article (what does this mean?). If published, this will include your full peer review and any attached files.

Reviewer #1: No

---

## [Author Response · Author response to Decision Letter 0]

25 Mar 2021

Comment from the reviewer：

Reviewer #1: This is an interesting article regarding the effectiveness of negative pressure barrier devices in providing air clearance to prevent aerosol transmission.

The topic is promising

Our response:

We are very pleased that you found our study interesting and informative. We have revised the manuscript thoroughly following your valuable suggestions. We hope the revised manuscript is now satisfactory.

Reviewer’s comment:

- The whole manuscript should be edited for English-language usage

Our response:

We apologize for the language in the manuscript. The document has now been proofread by an professional English language editor.

Reviewer’s comment:

- In the introduction the authors must describe the main aim avoiding too detailed descriptions that should be reported in the methods (primary and secondary etc.)

Our response:

Dear reviewer, we appreciate your valuable suggestion and adjusted the paragraph following your comment. We removed the part at the end of the introduction, which was more suitable in the study design and setting to improve the flow of the context.

The paragraph: ”Atomized glycerol was selected as a tracer gas and poly alpha olefins (PAO, diameter, 0.5–0.7 μm) was selected for aerosol concentration detection, based on previously published aerosol studies.24-26 The tracer gas with such a diameter was demonstrated to be best correlated with the aerosol droplet nuclei movement in an enclosed space.27 Our primary goal was to evaluate the effectiveness of each aerosol clearing device (AB, plastic aerosol bag, wall suction, and HEPA filter evacuator). Our secondary goal was to……” was removed from the introduction.

Reviewer’s comment:

- Is there a study period?

Our response:

Thank you for your query. This simulation experiment was conducted during 2020. The study period was April 20th to August 26th 2020. 

Reviewer’s comment:

- In the introduction the authors give too many references to describe a concept (2-10). Should be reduced by adding the following guidelines on the use of laparoscopy in surgery (and the connection with aerosolization)

The risk of COVID-19 transmission by laparoscopic smoke may be lower than for laparotomy: a narrative review. Surg Endosc. 2020 Aug;34(8):3298-3305. doi: 10.1007/s00464-020-07652-y

Italian society of colorectal surgery recommendations for good clinical practice in colorectal surgery during the novel coronavirus pandemic. Tech Coloproctol. 2020 Jun;24(6):501-505. doi: 10.1007/s10151-020-02209-6

A Low-cost, Safe, and Effective Method for Smoke Evacuation in Laparoscopic Surgery for Suspected Coronavirus Patients. Ann Surg. 2020 Jul;272(1):e7-e8. doi: 10.1097/SLA.0000000000003965

Our response:

We appreciate your valuable comments and have revised our references for each concept in this regard. Furthermore, three important studies that link the procedures and aerosolization have been added to the Reference list.

Reviewer’s comment:

- Were any patients involved? How many?

Our response:

Since our practice acquired a negative-pressure barrier device on January 19th 2021, a total of 32 patients have received tracheal intubation using the negative-pressure device.

Reviewer’s comment:

- The authors must clarify the methodology of the study

Our response:

Dear reviewer, we appreciate your important suggestion and had clarify the experiment setting and methodology of the study in the study design and settings to make sure the readers can understand the flow field environment for aerosol visualization and the process for aerosol concentration detection.

---

## [Decision Letter · Decision Letter 1]

30 Mar 2021

PONE-D-21-06215R1

Comparing the effectiveness of negative-pressure barrier devices in providing air clearance to prevent aerosol transmission

PLOS ONE

Dear Dr. Chen,

Thank you for submitting your manuscript to PLOS ONE. After careful consideration, we feel that it has merit but does not fully meet PLOS ONE’s publication criteria as it currently stands. Therefore, we invite you to submit a revised version of the manuscript that addresses the points raised during the review process.

ACADEMIC EDITOR:

Minor revision may be still necessary.You can decide which  references are suitable for citation to improve the quality of your manuscript.

We look forward to receiving your revised manuscript.

Kind regards,

Jianguo Wang, PhD

Academic Editor

PLOS ONE

Journal Requirements:

Reviewers' comments:

Reviewer's Responses to Questions

**Comments to the Author**

1. If the authors have adequately addressed your comments raised in a previous round of review and you feel that this manuscript is now acceptable for publication, you may indicate that here to bypass the “Comments to the Author” section, enter your conflict of interest statement in the “Confidential to Editor” section, and submit your "Accept" recommendation.

Reviewer #1: All comments have been addressed

2. Is the manuscript technically sound, and do the data support the conclusions?

Reviewer #1: Yes

3. Has the statistical analysis been performed appropriately and rigorously? 

Reviewer #1: Yes

4. Have the authors made all data underlying the findings in their manuscript fully available?

Reviewer #1: Yes

5. Is the manuscript presented in an intelligible fashion and written in standard English?

Reviewer #1: Yes

6. Review Comments to the Author

Reviewer #1: Congratulations to the authors for improving the manuscript.

I strongly suggest the following references regarding the use of the preventing systems for COVID-19 in surgery

The risk of COVID-19 transmission by laparoscopic smoke may be lower than for laparotomy: a narrative review. Surg Endosc. 2020 Aug;34(8):3298-3305. doi: 10.1007/s00464-020-07652-y

Italian society of colorectal surgery recommendations for good clinical practice in colorectal surgery during the novel coronavirus pandemic. Tech Coloproctol. 2020 Jun;24(6):501-505. doi: 10.1007/s10151-020-02209-6

A Low-cost, Safe, and Effective Method for Smoke Evacuation in Laparoscopic Surgery for Suspected Coronavirus Patients. Ann Surg. 2020 Jul;272(1):e7-e8. doi: 10.1097/SLA.0000000000003965

7. PLOS authors have the option to publish the peer review history of their article (what does this mean?). If published, this will include your full peer review and any attached files.

Reviewer #1: No

---

## [Author Response · Author response to Decision Letter 1]

30 Mar 2021

Comment from the reviewer：

Reviewer’s comment:

- In the introduction the authors give too many references to describe a concept (2-10). Should be reduced by adding the following guidelines on the use of laparoscopy in surgery (and the connection with aerosolization)

The risk of COVID-19 transmission by laparoscopic smoke may be lower than for laparotomy: a narrative review. Surg Endosc. 2020 Aug;34(8):3298-3305. doi: 10.1007/s00464-020-07652-y

A Low-cost, Safe, and Effective Method for Smoke Evacuation in Laparoscopic Surgery for Suspected Coronavirus Patients. Ann Surg. 2020 Jul;272(1):e7-e8. doi: 10.1097/SLA.0000000000003965

Our response:

We appreciate your valuable comments and have revised our references for each concept in this regard. Furthermore, these important studies that link the procedures and aerosolization have been added to the Reference list (10 and 20).

---

## [Decision Letter · Decision Letter 2]

5 Apr 2021

Comparing the effectiveness of negative-pressure barrier devices in providing air clearance to prevent aerosol transmission

PONE-D-21-06215R2

Dear Dr. Chen,

We’re pleased to inform you that your manuscript has been judged scientifically suitable for publication and will be formally accepted for publication once it meets all outstanding technical requirements.

Kind regards,

Jianguo Wang, PhD

Academic Editor

PLOS ONE

Additional Editor Comments (optional):

Reviewers' comments:

Reviewer's Responses to Questions

**Comments to the Author**

1. If the authors have adequately addressed your comments raised in a previous round of review and you feel that this manuscript is now acceptable for publication, you may indicate that here to bypass the “Comments to the Author” section, enter your conflict of interest statement in the “Confidential to Editor” section, and submit your "Accept" recommendation.

Reviewer #1: All comments have been addressed

2. Is the manuscript technically sound, and do the data support the conclusions?

Reviewer #1: Yes

3. Has the statistical analysis been performed appropriately and rigorously? 

Reviewer #1: Yes

4. Have the authors made all data underlying the findings in their manuscript fully available?

Reviewer #1: Yes

5. Is the manuscript presented in an intelligible fashion and written in standard English?

Reviewer #1: Yes

6. Review Comments to the Author

Reviewer #1: The authors have greatly improved the manuscript

I'm satisfied with the changes made

It can be considered for publication

7. PLOS authors have the option to publish the peer review history of their article (what does this mean?). If published, this will include your full peer review and any attached files.

Reviewer #1: No

---

## [Editor Report · Acceptance letter]

7 Apr 2021

PONE-D-21-06215R2 

Comparing the effectiveness of negative-pressure barrier devices in providing air clearance to prevent aerosol transmission 

Dear Dr. Chen:

I'm pleased to inform you that your manuscript has been deemed suitable for publication in PLOS ONE. Congratulations! Your manuscript is now with our production department. 

Kind regards, 

on behalf of

Dr. Jianguo Wang 

Academic Editor

PLOS ONE